# Recovery of Degraded Areas through Technosols and Mineral Nanoparticles: A Review

**Janaína Oliveira Gonçalves, Carolina Moreno Fruto, Mauricio Jaraba Barranco, Marcos Leandro Silva Oliveira and Claudete Gindri Ramos ***

Department of Civil and Environmental, Universidad de la Costa, Calle 58 #55-66, Barranquilla 080002, Colombia; joliveir@cuc.edu.co (J.O.G.); cmoreno9@cuc.edu.co (C.M.F.); mjaraba@cuc.edu.co (M.J.B.); msilva@cuc.edu.co (M.L.S.O.)
*   Correspondence: cgindri@cuc.edu.co

**Abstract:** Anthropogenic sources such as urban and agricultural runoff, fossil fuel combustion, domestic and industrial wastewater effluents, and atmospheric deposition generate large volumes of nutrient-rich organic and inorganic waste. In their original state under subsurface conditions, they can be inert and thermodynamically stable, although when some of their components are exposed to surface conditions, they undergo great physicochemical and mineralogical transformations, thereby mobilizing their constituents, which often end up contaminating the environment. These residues can be used in the production of technosols as agricultural inputs and the recovery of degraded areas. Technosol is defined as artificial soil made from organic and inorganic waste, capable of performing environmental and productive functions in a similar way to natural ones. This study presents results of international research on the use of technosol to increase soil fertility levels and recover degraded areas in some countries. The conclusions of the various studies served to expand the field of applicability of this line of research on technosols in contaminated spaces. The review indicated very promising results that support the sustainability of our ecosystem, and the improvement achieved with this procedure in soils is comparable to the hybridization and selection of plants that agriculture has performed for centuries to obtain better harvests. Thus, the use of a technosol presupposes a much faster recovery without the need for any other type of intervention.

**Keywords:** technosol; artificial soil; sustainability; solid waste; degraded soil recovery; clean production

## 1. Introduction

There is increasing awareness in the search for clean technologies and reuse of waste of the need for less social and environmental impact. Industries, in addition to worrying about production, cost, and efficiency, now explore sustainable technologies, mainly due to the COVID-19 pandemic and the new challenges and impacts that have arisen for the development of renewable technologies [1,2]. Therefore, environmental sustainability demands a collective effort from governments, companies, and the population in order to mature and follow important paths, especially with regard to technology. With this, a greater use of clean energy sources and renewable products has been sought, reducing waste generation and combating the pollution of water bodies and the soil [3–6]. These applied measures prevent further environmental damage and initiate favorable management and remediation planning due to the industrial adversity related to the environment. The rehabilitation of anthropic soil is one of the environmental impacts that have been of increasing concern, as its formation is a very slow, long-term process that may take years or millions of years to form [7,8].

Anthropogenic activities directly affect soil from the deposition of tailings and/or leaching of compounds, these interfere with acidity and also cause soil degradation through physical, chemical and biological transformations [6]. The mining activities are which are reflected in changes in the soil construction processes, several complex problems which

make even the attempt of recovery difficult. Normally, mined soils are infertile and develop as acidic soil, have low nutrient availability, and reduced organic materials [7–9]. To this end, studies come to be accomplished with the purpose of disposing of this waste before it becomes an environmental problem. Firpo et al. [10] proposed the use charcoal residues in a plant growth substrate, which is called technosol, which was combined with other residues such as rice husk ash, steel slag, and sewage sludge, resulting in a technosol capable of properly promoting plant growth and accelerate revegetation [10]. Guerrón et al. [11] verified the concentration of heavy metals from mine tailings in Technosols. The authors use a composite of nanoparticles, extracted from iron-stabilized orange peel. The retention of heavy metals reached 90% under optimal conditions, with a short treatment time, indicating promising results because it is the rich source of bioactive phenolic compounds and also because they corroborate the economic and environmental point of view.

Technosol is defined as artificial soils derived from anthropogenic sources, where their characteristics are dependent on the original properties and pedogenesis [12]. Technosols must have the ability to meet the basic needs of the soil, through the availability of nutrients, favoring the biological activity of the soil and increasing the soil water storage capacity. Artificial soils can be prepared from waste and serve as a substrate to recover compromised soils and assist in the development of vegetation [13,14]. The research on the use of Technosols in the recovery of degraded areas is still recent being that the first study was not the year 2007, is a topic that has not yet been addressed, but the studies that present very satisfactory results, combined with new sustainable technologies.

The work will indicate news sustainable technologies to corroborate future research that aims for a much faster recovery of degraded soils without the need for any other type of intervention, thus directing towards a rehabilitation system that performs the functions of self-sustainability, which fulfills the environmental and productive functions of soils. The receiving system becomes, as a result, less contaminated, preventing future damage and increasing ecosystem conservation. Therefore, numerous studies in the scientific literature demonstrate the potential of Technosols from different residues and their efficiency when compared to natural soils.

## 2. Technosols Regulations

Technosols are widely used in Spain. The pioneering regulation worldwide on the production of technosols as a waste recovery process was published in Galicia 2005, and modified in 2008 [15], with the following objectives: (i) Contribute to the simultaneous solution of waste management problems, through its recovery, and the recovery of degraded or contaminated soils at affordable and environmentally appropriate costs; (ii) Eliminate or strongly reduce the impacts of waste in the most sensitive systems such as water, air, and biota; (iii) Stabilize carbon in soils and biomass; (iv) Recycle nitrogen, phosphorus, potassium, and other macro and micronutrients, as well as materials that have useful components and properties; (v) Fulfill the productive and environmental functions in accordance with the European Soil Protection Strategy, which recommends the application of exogenous organic matter providing that it is of good quality and is used in accordance with good practices, considering the needs of the soil, its naturalness, and type of use, as well as the weather conditions.

The correct application of exogenous organic matter leads to the improvement of the physical, chemical, and biotic properties of soils, which are the increase in biological activity, biodiversity, better aggregation, and porosity, facilitating soil preparation and increasing its water-holding capacity. There is also an increase in the regulatory capacity of the soil in the face of various environmental impacts, which reinforces its water purification capacity while helping to combat climate change, reduce erosion and favor the recycling of elements and the valorization of waste [15].

According to Macías et al. [16], if the mechanisms of carbon stabilization in natural soils are known, as well as its genesis and evolution over time, depending on the geomorphological and climatic characteristics of the area, it is possible to develop artificial soils



that simulate the properties of Natural soils, by means of solidly structured mixtures of natural materials or of anthropic, organic, or inorganic origin, that respond to the definition subsequently agreed upon for Technosols. These Technosols prepared "*à la carte*" can be used as substitutes for degraded natural soils and/or contaminated by human action, to improve the previous environmental state [17].

The company Tratamientos Ecológicos Noroeste, located in Touro, uses Technosols for the restoration of the Touro mine that is produced by the Integrated Environmental Authorization, with the number of Resolutions and code: 2014-IPPC-56-283, and instead of the number of registration in the registry of producers and/or waste managers in the recovery activity that consists of the production of artificial materials: SC-I-IPPC-XV-00056 [18]. It improves the environmental and productive functions of soils, which allows for its biological activity, rebuilding biodiversity and food chains, the landscape, and plant productivity.

There are several studies and technical articles in this regard [19–21]. In summary, the characteristics sought in the Techniques used for restoration, according to the Centro de Valorización Ambiental de Touro, O Pino—VATOP [18] are: (i) Reducing potential, to minimize the oxidation rate of sulfides. Substances capable of reducing the redox potential below the $Fe^{3+}$ stability field and that consume oxygen are suitable because they slow down the oxidation of sulfides, thereby reducing the production of acidity. They are one such suitable solution among many others, including crushed remains from pruning and cleaning of the mountains, sewage sludge, food waste, elemental Fe, peat, good quality compost, and all materials rich in labile and easily degradable organic matter; (ii) Anti-acidifying agents, to neutralize the acidity produced, raise the pH so that the metals precipitate and favor the adsorption of sulfates and metals; (iii) Buffering agents, to control pH variations. Keeping the pH above 3.5 eliminates the $Fe^{3+}$, which drastically reduces the formation of acidity. If it is also possible to maintain a pH higher than 5.5. aquatic life, including fish populations, will be allowed in the waters coming out of the mine. The use of finely ground agricultural limestone has a rapid response, but short duration, as has been verified in repeated tests in the *As Pontes* mine, where amounts of up to 100,000 kg ha$^{-1}$ of $CaCO_3$ have only kept the pH high during the months of summer is quickly acidified after the first rains of autumn; (iv) Sulfate adsorbents, to reduce their mobility and that of metals. Heavy metal adsorption capacity. The soluble forms of heavy metals are highly toxic; hence, their fixation is an obligatory objective in the recovery of aquatic systems. In addition to the mechanisms of precipitation or co-precipitation, based on raising the pH, metal retention processes by surface adsorption are suitable. The materials can fix metals, but so do others such as peat, humid soils, residues from the production of fungi, etc; (v) Rich in nutrients, eutrophics, to encourage biotic activity and the biodiversity of organisms, increase plant production and the development of the trophic chain, as well as to provide necromass that makes recovery systems self-sustaining. Materials that contain significant amounts of assimilable forms of the main elements for feeding plants and soil microorganisms (C, N, P, and K), as well as secondary ones (Mg, Ca), are essential for the recovery of the biotic cycles. In this sense, in addition to synthetic fertilizers, organic fertilizers (manure, slurry, compost), plant residues (especially legumes) and agri-food residues are important.

The waste used in the production of technosols is sludge from urban wastewater treatment plants (WWTP) and agro-industry, livestock waste, plant biomass, other by-products such as ash, waste from construction, and civil demolition, sterile, mining. These technosols can be applied in the restoration of degraded soils, in agriculture, and in landscaping. This technology presents a sustainable alternative to replace soluble fertilizers, reduce the use of pesticides, restore degraded areas, and provide an environmentally sound destination for both urban and industrial organic and inorganic waste.

## 3. Technology and Sustainability

Technosols have appeared as a new technology to recover the post-mining scenario, but their efficiency depends on their ability to guarantee the properties of the soil. Tech-

nosols according to Food and Agriculture Organization—FAO must contain at least 20% of technogenic materials in the first 100 cm of the soil surface, either in the continuous rock or in the cemented layer [22], as illustrated in Figure 1.

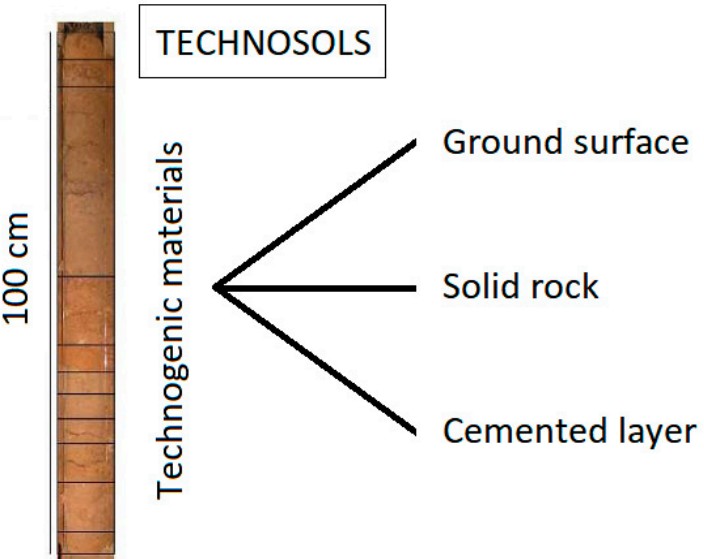

**Figure 1.** Outline definition of the term "Technosols".

The search for the use of technosols due to soil recovery is still considered recent, due to the beginning of the searches in 2007 (the first record found), and, until 2022, only 422 papers have been published in the form of scientific articles in Scopus, with 7 articles having already been published in the year 2022, so the topic is still considered "new". The evolution of publications can be seen in Figure 2.

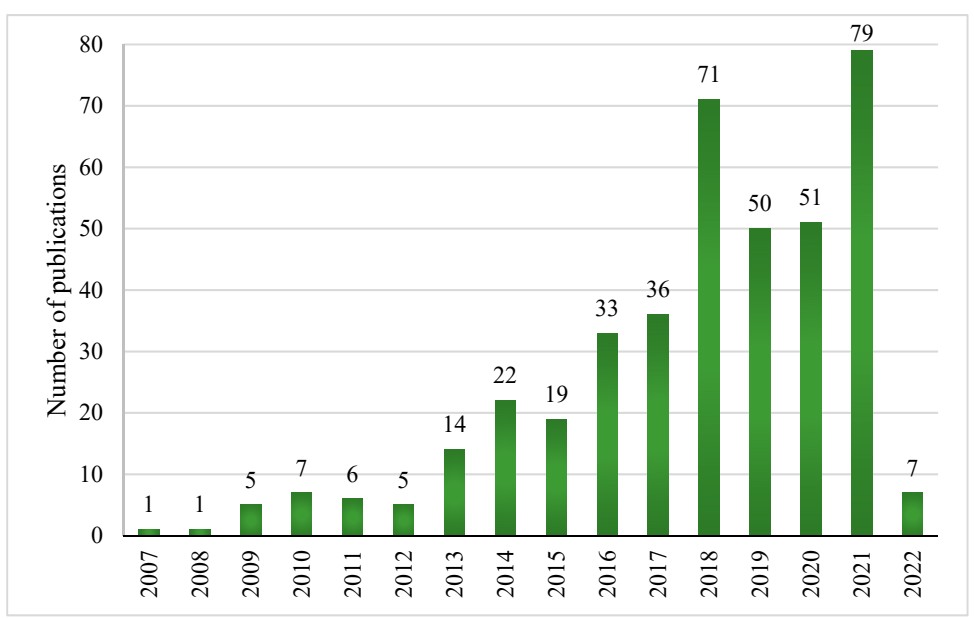

**Figure 2.** Chronological distribution of Technosols research publications in Scopus (2007–2022).

As is highlighted here, countries such as France, Spain, Germany, Russia, countries in the northern hemisphere, stand out for the largest number of publications on the subject. The largest number of publications were identified for the year 2018, with 71 articles, followed by the year 2021 with 79 articles on the topic. Therefore, this work indicates the

best sustainable technologies that were duly selected to support the preservation of our natural resources.

In total, 170 authors published their Technosols-related papers and Schwartz, C. was the most productive author who published 22 articles (see Table 1).

**Table 1.** The top 16 most productive authors with publications more than 7.

| Name | Total Publication | Name | Total Publication |
|---|---|---|---|
| Schwartz, C. | 22 | Banov, M. | 9 |
| Morel, J.L. | 14 | Charzyński, P. | 9 |
| Séré, G. | 14 | Atanassova, I. | 8 |
| Watteau, F. | 14 | Greinert, A. | 8 |
| Macías, F. | 12 | Nehls, T. | 8 |
| Pietrzykowski, M. | 12 | Woś, B. | 8 |
| Uzarowicz, Ł. | 11 | Abreu, M.M. | 7 |
| Minkina, T. | 10 | Ouvrard, S. | 7 |

With respect to the source journals, these 422 publications were published across 131 journals. Table 2 shows the 16 most productive journals, accounting for 47.63% of the total publications. Journal of Soils and Sediments was the most productive journal with 52 articles, accounting for 12.32% of the total publications.

**Table 2.** The 12 most productive Scopus journals for Technosols research from 2007 to 2022.

| Journals | Number of Publications |
|---|---|
| Journal of Soils and Sediments | 52 |
| Geoderma | 31 |
| Eurasian Soil Science | 22 |
| Catena | 17 |
| Soil Science Annual | 14 |
| Ecological Engineering | 11 |
| Environmental Geochemistry and Health | 11 |
| Chemosphere | 10 |
| Journal of Environmental Management | 9 |
| Applied Soil Ecology | 8 |
| Science of the Total Environment | 8 |
| Springer Geography | 8 |

The applicability of technosols makes it possible to increase production or production capacity in a defined area, for example, the use of technosols for greater water retention, to regulate its vegetative period and improve plant productivity. In other words, to understand the proper treatment that an area requires, the study and prior sampling must first be carried out in the laboratory and then pilot technosols that are found to be tested in small quantities in the area should be developed. After this test and after defining which is the most suitable according to need or the types of pollutants to be treated, it should be applied throughout the study area, leading to industrial applicability. The manufacture of a technosol can take several months, but after its application, the respective monitoring is carried out, such as the periodic control of pH and the presence of contaminants. An alarming problem among industrial sectors around the world is soil contamination from potentially toxic elements (PTE). Potentially toxic elements bring countless harm to the environment due to their complex processes with the soil. For example, the adsorption of heavy metals in the soil and its permanence for long years leads to change in soil pH, infertility, and irreversible losses. Therefore, there are currently several studies based on the use of new resources such as waste, biodegradable products, thereby leading to an increase in the use of technosols to try to alleviate this problem.

The study in relation to Technosols goes beyond a response through its remediation in soils, factors such as degradation, alteration of organic matter, porosity, handling, mixing,

sealing, and excavation, are also relevant when using these artificial soils. Therefore, these aspects must be considered when using Technosols to identify and understand the processes of pedogenesis.

The search for sustainable alternatives has increased due to concern for the environment and human health. There are several ways in which these PTE's can present such as in the air, water, and soil, which can be inhaled, consumed indirectly by agricultural production, or by direct poisoning [23]. With this, new challenges arise with the principle of innovation, technology, and sustainability in the mining area. In this manuscript, there are several alternatives, ideas and guidelines to support the post-ore treatment and, therefore, facilitate the search for efficient, low-cost methods to use waste that are unusable.

One of the biggest concerns these days is mining, as it is proven that we cannot live without mining in our daily lives. From our home, transport, food, tools, utensils, everything relies on mining, but there are alarming consequences of this anthropogenic activity. Among them, the destruction of soils that take thousands of years to form together is the main issue, resulting from the accumulation of polluting materials that are harmful to pedogenesis. Therefore, studies such as this are needed in the search for adequate treatments to reverse this great impact. In addition to needing a thorough investigation and recognition, we also need a prolonged period of recovery of these soils. The formation of soil is already a very extensive process and recovery can also be, some studies indicate 42 years to have the desired return, such as greater water retention capacity, increased porosity, and soil horizons, with that generating better fertility [24]. However, other studies have shown promising results for periods of 6 to 18 months of recovery [25,26]. With that, this work will indicate different recovery periods, with different residues that would be discarded incorrectly, and that nowadays, can be combined with Technosols for its best destination. Besides having an adequate end, these will have promising effects for soils harmed by anthropogenic activities.

Coal mining residues present in soils necessitate a recovery process because they cause negative impacts such as the leaching of toxic compounds, the disappearance of bio habitats, water contamination, and soil degradation processes. Thus, alternatives are necessary for us to convert this problem from stimulating soil formation processes, as well as providing biological activity on the affected surface. The technique commonly used was to employ an impermeable layer over these residues and immediately a new layer of soil, thereby favoring the development of vegetation and avoiding degradation processes caused by external factors, water, and wind, which cause soil erosion. This technique is not very effective because the toxic residues are "buried" but will still be present. With this, new studies are being carried out to reuse these residues. Firpo et al. [10], for example, carried out their study with the purpose of using their coal residues as a technosol [10]. Mining waste, along with other types of waste (steel slag, sewage sludge, and rice husk ash) served as a substrate for plant growth. The authors claim that the technosol was highly efficient in soil rehabilitation and made the soil fertile. Furthermore, the authors describe that the amount of metal found in the soil is within the proper range, which is of great relevance for the study since the biggest problem in the disposal of coal tailings is the presence of highly toxic metals.

The mining sector is more submissive to waste recycling and its remediation, as well as to the treatment of contaminated soil. Slukovskaya et al. [27] used technosol engineering in the mining tailings (carbonatite and serpentinite-magnesite) from the copper-nickel plant in the Murmansk region, Russia [27]. The authors used grasses in degraded soil covered with hydroponic vermiculite and verified the efficiency of the technosols after 7 years of sowing the grass. The two residues had high levels of calcium and manganese minerals; moreover, the authors reported results that corroborate the pedogenesis and rehabilitation of problematic soil from anthropogenic activities.

An important factor in soil pedogenesis and recomposition is the amount of organic matter (OM) that is present, PTE acts as an inhibitor in the seasonality of humic acid molecules, directly affecting organic matter and thus the biochemical processes of the soil.

To estimate soil degradation as a function of PTE, enzymes have been applied to investigate this decrease in biological processes because they are extremely sensitive to changes in the soil. Zamulina et al. [9] evaluated the activities of two enzymes under the effect of Zn and Cu contamination, as a function of the water-soluble organic matter present in Spolic Technosols [9]. The authors indicated that the options of contaminants that do not present soil directly affect the degree of biological activity, thereby affecting organic matter, pH, and the proportion of carbonate in the soil. Another study focused on investigating the importance of OM present in the soil and carried out by Vidal-Beaudet et al. [28] concerned urban waste and by-product mixtures to designate soil fertility. The authors selected two plants *Lolium perenne* (ryegrass) and *Brassica napus* (rape) and verified the phosphorus content in the technosol mineralization, moreover, reported the early development of pedogenesis processes.

The "state" of the soil indicates a range of biochemical factors, such as geochemical syntheses, the amount of OM and nutrient fluctuations, all of which are directly affected by mining activities. Therefore, the evaluation of the soil properties is of great importance so that remediation can be carried out. In tropical climate regions, for example, the effect of mine fires should be evaluated, as well as the location of the soil moisture content, which favors the deposition of salts and the concentration of other minerals on the surface; therefore, before applying the technosol, these characteristics they must clearly have a favorable and effective remedy. Ahirwal et al. [29] state that the age of revegetation also influences previous changes in the soil, as a function of decomposition and associated activities such as plants [29]. The authors reported the higher the age of revegetation using a technosol, the greater the capacity to retain water in the soil, and with this there is an increase in the moisture content, favoring the formation of biochemical processes and with that, an enhanced improvement in infertility from the soil.

The research is focused on methods of bioremediation, phytoremediation, and biotechnological methods to reduce the impact generated on the ecosystem. One of these sustainable technologies that are being employed is the use of nutrient-rich sediments to reduce the acidity of metals and consequently increase soil pH. Sediments can be used for urban afforestation, land reclamation, and crops [30,31]. Sediments from aquatic organisms for plant cultivation are being used as technosol formers. Cortinhas et al. [31] addressed the use of two irrigation techniques (use of saltwater and deionized water) in carrying out the experiments, as they emphasize that aquaculture is carried out in brackish conditions. With this, the authors concluded that the technosols formed from these sediments tolerate saline conditions and can be supplied in agricultural products for human consumption and for use by the pharmaceutical industries.

The disturbance induced by anthropogenic activities has a global dimension, whether by the accumulation of garbage (urban or industrial), resource ablation, and urbanization. Recovery processes range from modifying the soil cover, in situ treatments, or even an ex situ treatment that consists of excavating the contaminated soil and applying the treatment in another location. The choice of these methods originates from previous studies and from the availability and amount of soil to be treated. Usually, mining activities generate high amounts of residues and tailings. Because these treatments are carried out in the affected location, and because the residues can be exposed to air and be carried by the wind when being transported the application of technosols plays an important role in the correction of these degraded areas (Figure 3).

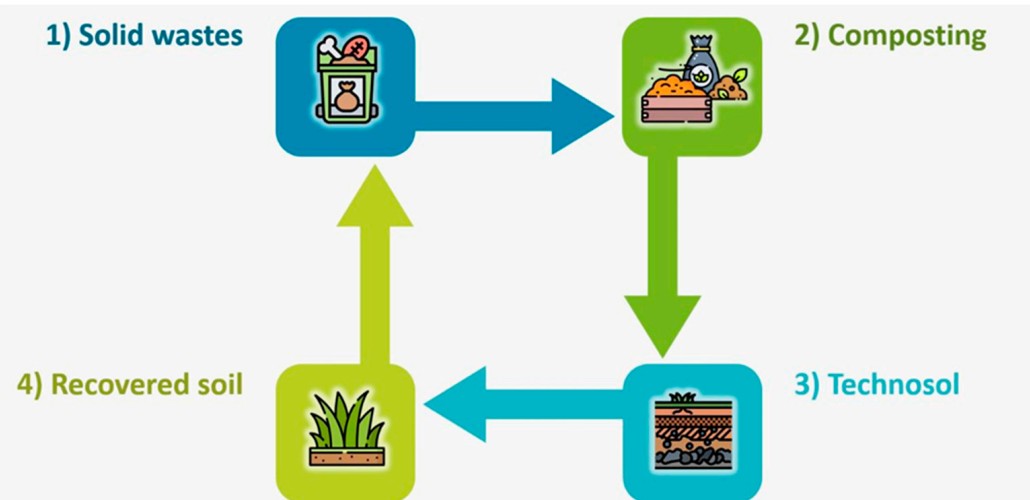

**Figure 3.** Summary of the Technosols system.

In their study, Ruiz et al. [32] evaluated the effectiveness of using technosols under sugarcane and pasture in terms of the recovery of a limestone mine, the results were very promising in terms of the quality of these soils when compared to native soil, that is, technosols recovered their properties and higher mineral values than in native soils. Therefore, limestone residues restore the basic functions of the soil in a reduced application time.

Uzarowicz et al. [33] carried out a study verifying the efficiency of a technosol after 45 years, in which they related the effect of biological activities as a function of the basic properties of soil. The authors reported that pH and salinity are directly related to biological activities and concluded that the use of this technosol was effective in soil recovery in the vicinity of the former uranium iron sulfide mine in the village of Rudki (Holy Cross Mts., Poland).

The reuse of mining waste together with other sectors, such as civil construction, has expanded the possibilities of its application. The use of technosols formed from waste recycling is a promising and sustainable alternative since the soil supports the biodiversity of organisms different to the natural environment [34]. The reuse of civil construction waste is still low, but its disposal can generate environmental, aesthetic, and public health problems, and overload municipal public cleaning systems. The use of these compounds as technosols indicates promising results as they provide the chemical elements necessary to nourish the plants and enable the development of vegetation with a quality comparable or superior to that of natural soils.

Pruvost et al. [35] proposed the mixtures of excavated soil horizons, crushed concrete, and green waste compost to assess the behavior of macrofauna through soil fertility, as they are fundamental in stimulating productivity. The study was carried out over a very short period (3 years) and showed significant results, which reveals that the responses obtained were very prominent in the formation of technosols from urban waste, making it one more possibility to reestablish the soil and consequently the fauna. Foti et al. [36] and Pruvost et al. [35] recommend the use of green urban waste and construction waste as an alternative to introducing this type of soil instead of extracting natural soils for gardening and landscaping purposes.

Mine soils are normally considered practically irremediable due to the presence of forests which inhibit the formation of soil and the development of vegetation. Therefore, studies aim to seek an appropriate treatment to reduce the impact generated by these compounds. Phytoremediation combined with the use of some technosol has played an effective role in the correction of degraded soils. According to Benhabylès et al. [37] phytore-remediation is the process most used in large, contaminated areas and the use of technosol with compound additives and biochar as a fertilizer is very effective, not only as a nutrient, and acts as a barrier to prevent erosion of the ground. Lebrun et al. propose that

the use of technosol with biochar reduces soil acidity, immobilizes metals, increases the OM content, and reduces the costs of applying biochar. In their study, the authors used a plant, the linen [38]. The results indicated improvements in plant development with controlled doses of biochar.

Remediation methods are particularly effective in the treatment of areas degraded by mining soils due to their cost-effectiveness and effectiveness. An indicator of soil quality and soil efficiency can only be verified after 6 years because soil aggregates that are suitable for OM retention in these reclaimed areas are observed [39,40]. According to Halecki et al. [41] 8 years were needed to state that post-mining soil recovery using technosols from sewage sludge enriched with Carbocrash substrate was effective. The authors presented promising results in restoring the physical and chemical properties of the soil, stating that mineral fertilizers can be replaced by sewage sludge while maintaining their effectiveness.

There are several techniques that have been applied in the recovery of degraded soils, among which the most used are the implementation of technosols and the use of biochar. Biochar is obtained from the carbonization of biomass and emerged with the aim of replacing the use of chemical fertilizers. Therefore, studies show promising results such as those presented by Forhán et al. [42], in which they compare the efficiencies of these two recovery methods together in the soil, and in which the authors considered the variation in pH, amount of nitrogen, and total carbon, obtaining nutrients and exchange cationic. In addition, the nutrient fixation capacity from biochar was also evaluated to corroborate the purposes of technosols in the development of *Brassica juncea* L. The results presented were very significant and promising as the study was carried out over a period short time (11 months).

Another study that was also carried out over a short period and with satisfactory results was the comparison of the use of compost and technosol from residues, which were tested at different depths in the soil. These residues were composed of industrial aluminum tailings, from a purification plant, a cellulose company, and from agri-food industries. The compost was obtained from rabbit and horse manure, which were aggregated with pruning residues, fruits, and seaweed. The soil sample was taken from a depleted copper mine, and collected in a decantation pond that was formed from deposits of sulfide flotation during copper elaboration. The results showed that the soil treated with the compost had a more promising result in the reduction of PTE at depths of 0–15 and 15–30 cm, whereas at depths of 30–45 cm the results were the same. However, the authors point out that in 3 months a relevant value of the translocation factor for all metals in soils treated with technosol was already observed. Therefore, the study indicated that both the compound and technosol showed effective performance in the phytostabilization process [25].

Metals are the main cause of soil contamination among all inorganic pollutants; therefore most soil remediation methods use some sort of chemical agent. With this, the authors Nandillon et al. [43] established a strategy as an alternative method using phytoremediation with the germination of black poplar seeds combined with a technosol applied to a highly polluted soil. The work showed the correction of the soil and the germination of seeds in soil, which are unfavorable to any type of vegetation. They concluded the effectiveness in the revegetation and stabilization of Pb in the soil, compost was added to technosol. Therefore, more complex studies on this peculiarity of As are necessary so that there is an improvement in its stability in Technosols.

From these presented studies, it is possible to state that the behavior of a Technosol is similar to that of natural soil, whether in structuring, settlement, or subdivision, but when using artificial soil, we can be more specific and obtain a remeasurement response or faster immobilization and achieve the reuse of some waste that would otherwise be discarded incorrectly [44–47]. Therefore, this alternative is increasingly used in scientific investigations, and it is noteworthy that some complications such as the weathering of materials, formation of secondary materials, precipitation, and segregation may arise

from certain types of waste [48,49]. So, care in managing the use of a Technosol must be monitored, and followed up to avoid possible inconsistencies that may arise.

Currently, there are numerous clean technologies available to treat pollutants [1–3,50,51] and Technosols has been gaining prominence in the literature because they consider existing dysfunctions, reinforce the attenuation capacity of natural systems, and regulate the hydrogenic potential and the critical number of toxic pollutants present in the soil [52–55]. Therefore, they appear as an efficient alternative for ecosystem recovery as they are intentionally designed or modified by man to meet specific needs, such as waste management and storage, support for industrial activities, and mobilization [56–60].

## 4. Conclusions

Research on issues related to environmental sustainability becomes relevant as the exploitation of natural resources increases. The biggest challenges for future lines of research is to reduce energy consumption and increase the use of renewable sources, analyze the social sustainability, especially of mining activities, and incorporate clean production into the development of new environments after mines have close. With that, this work brings new perspectives for the analysis and integral evaluation of innovations and technologies using technosols that aim to make industrial activity more sustainable, mainly valuing contributions in the environmental, economic, and social domains. Technosols presents a sustainable and promising path in terms of its technology, which is in continuous development and evolution. Therefore, it is necessary to continuously monitor the behavior and evolution of this soil, to determine its scope and adequate applicability. The conclusions of the various studies served to broaden the field of action of this line of research on technosols in contaminated spaces. The review indicated very promising results that corroborate the sustainability of our ecosystem, and the improvement achieved with this procedure in soils is comparable to the hybridization and selection of plants that agriculture has performed for centuries to achieve better harvests. Thus, the use of a technosols assumes a much faster recovery without the need for any other type of intervention. In addition, we emphasize here the importance of future research on the influence of some specific nanoparticles and their effect on the use of Technosols.

**Author Contributions:** Conceptualization, J.O.G. and C.G.R.; methodology, J.O.G.; validation, C.M.F. and M.J.B.; investigation, J.O.G. and C.G.R.; resources, J.O.G. writing—original J.O.G. and C.G.R. draft preparation, C.M.F.; writing—review and editing, J.O.G. and C.G.R.; visualization, M.J.B.; supervision, C.G.R. and M.L.S.O.; project administration, C.G.R. and M.L.S.O. All authors have read and agreed to the published version of the manuscript.

**Funding:** This research received no external funding.

**Institutional Review Board Statement:** Not applicable.

**Informed Consent Statement:** Not applicable.

**Data Availability Statement:** Not applicable.

**Conflicts of Interest:** The authors declare no conflict of interest.

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
