# Peer review of "Recovery of Degraded Areas through Technosols and Mineral Nanoparticles: A Review"

_sustainability, doi:10.3390/su14020993_

Round 1

Reviewer 1 Report

This paper deals with a comprehensive analysis of the worldwide scientific literature on the recovery of degraded areas through Technosols and mineral nanoparticles. The paper indicated very promising results that support the sustainability of our ecosystem, and the improvement achieved with this procedure in soils is comparable to the hybridization and selection of plants that agriculture has been doing for centuries to obtain better harvests. Thus, the use of a technosol presupposes a much faster recovery without the need for any other type of intervention. This paper is interesting, however, it should be revised before publication. 
1.    In the Introduction section, the authors should mention some  clean technologies and reuse of waste so that there is a less social and environmental impact. The authors should refer to pollutant and some methods to recover/treat these pollutant as well as clean technologies for the future. The authors should refer to some recent publication Impacts of COVID-19 pandemic on the global energy system and the shift progress to renewable energy: Opportunities, challenges, and policy implications; Integrating renewable sources into energy system for smart city as a sagacious strategy towards clean and sustainable process; Scrap tire pyrolysis as a potential strategy for waste management pathway: a review; Endless story about the alarming reality of plastic waste in Vietnam; Heavy metal removal by biomass-derived carbon nanotubes as a greener environmental remediation: A comprehensive review; Microbial fuel cells for bioelectricity production from waste as sustainable prospect of future energy sector. 
2.    The authors need to clarify and explain the difference between the current study with the available literature, as well as the main contribution of the study to science. A novelty in the work was not properly highlighted and explained.
3.    The authors should add some other Figures to enrich their content. 
4.    The novelty of the work must be clearly addressed and discussed, compare your research with existing research findings, and highlight novelty, (compare your work with existing research findings and highlight novelty)
5.    The conclusion section is missing some perspective related to the future research work, quantifying main research findings.

Author Response

Dear editor:

Thank you for your useful comments of our manuscript. We have modified the manuscript accordingly, and each point cited by referees was revised. In the revised version, the modifications are in red.

Review Repor 1

  1.    In the Introduction section, the authors should mention some  clean technologies and reuse of waste so that there is a less social and environmental impact. The authors should refer to pollutant and some methods to recover/treat these pollutant as well as clean technologies for the future. The authors should refer to some recent publication Impacts of COVID-19 pandemic on the global energy system and the shift progress to renewable energy: Opportunities, challenges, and policy implications; Integrating renewable sources into energy system for smart city as a sagacious strategy towards clean and sustainable process; Scrap tire pyrolysis as a potential strategy for waste management pathway: a review; Endless story about the alarming reality of plastic waste in Vietnam; Heavy metal removal by biomass-derived carbon nanotubes as a greener environmental remediation: A comprehensive review; Microbial fuel cells for bioelectricity production from waste as sustainable prospect of future energy sector. 

Response: Thanks for the suggestion, it was modified in the manuscript and some more references were added due to the clean technologies used.

  1.    The authors need to clarify and explain the difference between the current study with the available literature, as well as the main contribution of the study to science. A novelty in the work was not properly highlighted and explained.

Response: Thanks for the suggestion, it was modified in manuscript

  1.    The authors should add some other Figures to enrich their content. 

Response: Thanks for the suggestion, it was added news Figures in manuscript

  1.    The novelty of the work must be clearly addressed and discussed, compare your research with existing research findings, and highlight novelty, (compare your work with existing research findings and highlight novelty)

Response: Sorry, we appreciate your comment and agree. The manuscript was changed, and a new discussion was added.

  1.    The conclusion section is missing some perspective related to the future research work, quantifying main research findings.

Response: Sorry, it was changed to the conclusion of the manuscript.

Reviewer 2 Report

The authors reviewed the very topical issue of Recovery of degraded areas through technosols and mineral nanoparticles, the attention to which is increasing every year.

The title should be changed, leaving only the content part of "Recovery of degraded areas through technosols and mineral nanoparticles"

The first sentence of the abstract needs to be adjusted by grouping pollution sources and excluding duplicate building types.

At the end, the formal fields must be removed or filled.

The work does not contain overview tables, which are a standard element of review articles, in which sources are grouped according to the variants of recovery of degraded areas considered in them and the key ideas and approaches of these works are highlighted.

Graph 1 shows the number of publications over the past 15 years, it is 383 papers, but the number of publications considered in this paper is only 34. This is very small for a review article. It is necessary to increase the number of considered works at least to 60, and better still more of them.

Author Response

Dear editor:

Thank you for your useful comments of our manuscript. We have modified the manuscript accordingly, and each point cited by referees was revised. In the revised version, the modifications are in red.

Review Repor 2

1.The title should be changed, leaving only the content part of "Recovery of degraded areas through technosols and mineral nanoparticles"

Response: Thanks for the suggestion, the title was modified

2.The first sentence of the abstract needs to be adjusted by grouping pollution sources and excluding duplicate building types.

Response: Thanks for the suggestion, it was modified to “Anthropogenic sources such as urban and agricultural runoff, fossil fuel combustion, domestic and industrial wastewater effluents, and atmospheric deposition generate large volumes of nutrient-rich organic and inorganic waste.”

  1. At the end, the formal fields must be removed or filled.

Response: Thanks, the formal fields must be modified.

  1. The work does not contain overview tables, which are a standard element of review articles, in which sources are grouped according to the variants of recovery of degraded areas considered in them and the key ideas and approaches of these works are highlighted.

Response: Thanks for the suggestion, we´ve added a table in the manuscript

  1. Graph 1 shows the number of publications over the past 15 years, it is 383 papers, but the number of publications considered in this paper is only 34. This is very small for a review article. It is necessary to increase the number of considered works at least to 60, and better still more of them

Response: Thanks for the suggestion, we've added more discussion and more references to the manuscript.

Round 2

Reviewer 1 Report

The authors have addressed very well my requirement. Now, it could be accepted for publication. 

Reviewer 2 Report

The authors have made the necessary changes to the article. The article can be published in the journal.